# DELT: A Simple Diversity-driven `EarlyLate` Training for Dataset Distillation

## Abstract

Recent advances in dataset distillation have led to solutions in two main directions. The conventional *batch-to-batch* matching mechanism is ideal for small-scale datasets and includes bi-level optimization methods on models and syntheses, such as FRePo, RCIG, and RaT-BPTT, as well as other methods like distribution matching, gradient matching, and weight trajectory matching. Conversely, *batch-to-global* matching typifies decoupled methods, which are particularly advantageous for large-scale datasets. This approach has garnered substantial interest within the community, as seen in SRe$^2$L, G-VBSM, WMDD, and CDA. A primary challenge with the second approach is the lack of diversity among syntheses within each class since samples are optimized independently and the same global supervision signals are reused across different synthetic images. In this study, we propose a new `EarlyLate` training scheme to enhance the diversity of images in *batch-to-global* matching with less computation. Our approach is conceptually simple yet effective, it partitions predefined IPC samples into smaller subtasks and employs local optimizations to distill each subset into distributions from distinct phases, reducing the uniformity induced by the unified optimization process. These distilled images from the subtasks demonstrate effective generalization when applied to the entire task. We conducted extensive experiments on CIFAR, Tiny-ImageNet, ImageNet-1K, and its sub-datasets. Our empirical results demonstrate that the proposed approach significantly improves over previous state-of-the-art methods under various IPCs[1].

## 1   Introduction

In the era of large models and large datasets, dataset distillation has emerged as a crucial strategy to enhance training efficiency and make AI technologies more accessible and affordable for the general public. Previous approaches [1, 2, 3, 4, 5, 6, 7, 8, 9, 10] primarily employ a *batch-to-batch* matching technique, where information like features, gradients, and trajectories from a local original data batch are used to supervise and train a corresponding batch of generated data. This method's strength lies in its ability to capture fine-grained information from the original data, as each batch's supervision signals vary. However, the downside is the necessity to repeatedly

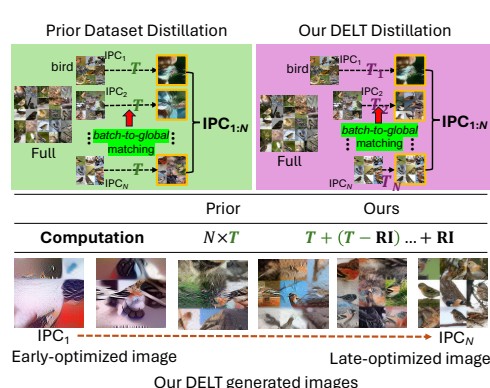

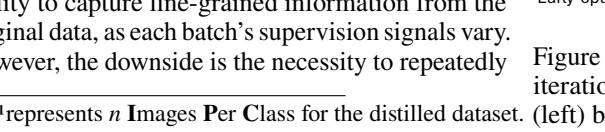

Figure 1: Distill datasets to IPC$_N$ requires $N * T$ iterations in traditional distillation processes (left) but fewer iteration processes (right).

---

[1]represents $n$ **I**mages **P**er **C**lass for the distilled dataset.

Submitted to 38th Conference on Neural Information Processing Systems (NeurIPS 2024). Do not distribute.

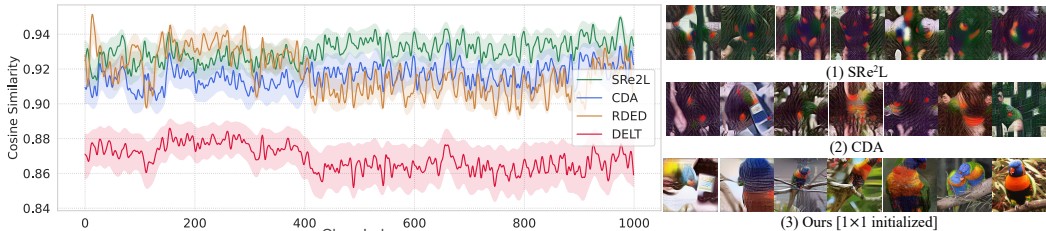

Figure 2: **Left**: Intra-class semantic cosine similarity after a pretrained ResNet-18 model on ImageNet-1K dataset, lower values are better. **Right**: Synthetic images from SRe²L, CDA and our DELT.

input both original and generated data for each training iteration, which significantly increases memory usage and computational costs. Recently, a new decoupled method [11, 12, 13] has been proposed to separate the model training and data synthesis, also it leverages the *batch-to-global* matching to avoid inputting original data during distilled data generation. This solution has demonstrated great advantage on large-scale datasets like ImageNet-1K [11, 14] and ImageNet-21K [12]. However, as shown in Fig. 2 right subfigure, a significant limitation of this method is its strategy of synthesizing each data point individually, where supervision is repetitively applied across various synthetic images. For instance, SRe²L[11] utilizes globally-counted layer-wise running means and variances from the pre-trained model for supervising different intra-class image synthesis. This methodology results in a pronounced lack of diversity within the same category of generated images.

To address this issue, previous studies such as G-VBSM [14] and RDED [15] have been conducted. Specifically, G-VBSM [14] introduces a framework that utilizes a diverse set of *local-match-global* matching signals derived from multiple backbones and statistical metrics, offering more precise and effective matching than the singular model. However, as the diversity of matching models grows, the overall complexity of the framework also increases, thus diminishing its conciseness. RDED [15] crops each original image into multiple patches and ranks these using realism scores generated by an observer model. Then it amalgamates every four chosen patches from previous stage into a single new image, maintaining the resolution of the original images, and produce IPC-numbered distilled images for each class. While RDED is effective for selecting and combining data, it does not enhance or optimize the visual content within the distilled dataset. Thus, the diversity and richness of information it encapsulates largely dependent on the distribution of the original dataset.

Our solution, termed the `EarlyLate` training scheme, is straightforward and also orthogonal to these prior methods: by initializing each image in the same category at a different starting point for optimization, we ensure that the final optimized results vary across images. We also use teacher-ranked real image patches to initialize the synthetic images. This prevents some images from being short-optimized and ensures they provide sufficient information. As shown in Fig. 1 of the computation comparison, our approach not only enhances intra-class diversity but also significantly reduces the computational load of the training process. Specifically, while conventional training requires $T$ optimization iterations per image or batch, in our `EarlyLate` scheme, the first image undergoes $T_1$ iterations (where $T_1 = T$). Subsequent batches are processed with progressively fewer iterations, such as $T_2$ ($T_2 = T_1 - RI$²) for the next set, and so forth. The iterations for the final batch are reduced to RI which is $1/j$ of the standard count (where typically $j = 4$ or 8), meaning the total number of optimization iterations required is just about 2/3 of prior *batch-to-global* matching methods, such as SRe²L and CDA. We further visualize the average cosine similarity between each sample of 50 IPCs with the associated cluster centroid within the same class on ImageNet-1K, as shown in Fig. 2 left subfigure, DELT shows significantly better diversity than other counterpart methods across all classes.

We perform extensive experiments on datasets of CIFAR-10, Tiny-ImageNet, ImageNet-1K and its subsets. On ImageNet-1K, our proposed approach achieves 66.1% under IPC 50 with ResNet-101, outperforming previous state-of-the-art RDED by 4.9%. On small-scale datasets of CIFAR-10, our approach also obtains 2.5% and 19.2% improvement over RDED and SRe²L using ResNet-101.

Our main contributions in this work are as follows:

- We propose a simple yet effective `EarlyLate` training scheme for dataset distillation to enhance the intra-class diversity of synthetic images from *batch-to-global* matching.

---

²RI is the number of round iterations and will be introduced in Sec. 4.3.

- We demonstrate empirically that the proposed method can generate optimized images at different distances from their initializations, to enlarge informativeness among generations.
- We conducted extensive experiments and ablations on various datasets across different scales to prove the effectiveness of the proposed approach[3].

## 2 Related Work

**Dataset Distillation.** Dataset distillation or condensation [1] focuses on creating a compact yet representative subset from a large original dataset. This enables more efficient model training while maintaining the ability to evaluate on the original test data distribution and achieve satisfactory performance. Previous works [1, 2, 3, 4, 5, 6, 7, 8, 9, 10] mainly designed how to better match the distribution between original data and generated data in a *batch-to-batch* manner, such as the distribution of features [6], gradients [2], or the model weight trajectories [4, 8]. The primary optimization method used is bi-level optimization [16, 17], which involves optimizing model parameters and updating images simultaneously. For instance, using gradient matching, the process can be formulated as to minimize the gradient distance:

$$\min_{\mathcal{S} \in \mathbb{R}^{N \times d}} D\left(\nabla_\theta \ell(\mathcal{S}; \theta), \nabla_\theta \ell(\mathcal{T}; \theta)\right) = D(\mathcal{S}, \mathcal{T}; \theta) \tag{1}$$

where the function $D(\cdot, \cdot)$ is defined as a distance metric such as MSE [18], $\theta$ denotes the model parameters, and $\nabla_\theta \ell(\cdot; \theta)$ represents the gradient, utilizing either the original dataset $\mathcal{T}$ or its synthetic version $\mathcal{S}$. $N$ is the number of $d$-dimensional synthetic data. During distillation, the synthetic dataset $\mathcal{S}$ and model $\theta$ are updated alternatively,

$$\mathcal{S} \leftarrow \mathcal{S} - \lambda \nabla_{\mathcal{S}} D(\mathcal{S}, \mathcal{T}; \theta), \quad \theta \leftarrow \theta - \eta \nabla_\theta \ell(\theta; \mathcal{S}), \tag{2}$$

where $\lambda$ and $\eta$ are learning rates designated for $\mathcal{S}$ and $\theta$, respectively.

*Batch-to-global* matching used in [11, 14, 12, 13] tracks the distribution of BN statistics derived from the original dataset for the local batch synthetic data, the formulation can be:

$$\min_{\mathcal{S} \in \mathbb{R}^{N \times d}} \left( \sum_l \left\| \mu_l(\mathcal{S}) - \mathbf{BN}_l^{\mathrm{RM}} \right\|_2 + \sum_l \left\| \sigma_l^2(\mathcal{S}) - \mathbf{BN}_l^{\mathrm{RV}} \right\|_2 \right) \tag{3}$$

where $l$ is the index of BN layer, $\mu_l(\mathcal{S})$ and $\sigma_l^2(\mathcal{S})$ are mean and variance. $\mathbf{BN}_l^{\mathrm{RM}}$ and $\mathbf{BN}_l^{\mathrm{RV}}$ are running mean and running variance in the pre-trained model at $l$-th layer, which are globally counted. Fig. 3 illustrates the difference of *batch-to-batch* and *batch-to-global* matching mechanisms, where $b$ represents a local batch in data $\mathcal{T}$ and $\mathcal{S}$.

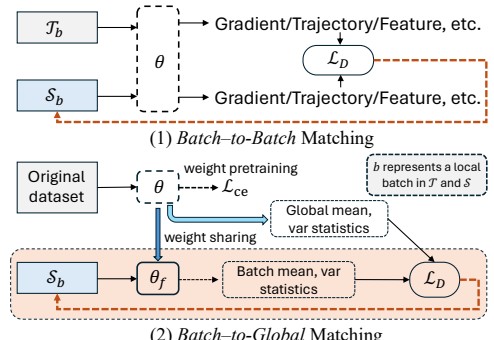

(1) *Batch–to–Batch* Matching

(2) *Batch–to–Global* Matching

Figure 3: *Batch–to–batch* vs. *batch-to-global* matching in dataset distillation. $\theta_f$ indicates weights are pretrained and frozen in this stage.

Moreover, for the recent advances of multi-stage dataset distillation methods, MDC [10] proposes to compress multiple condensation processes into a single one by including an adaptive subset loss on top of the basic condensation loss, so that to obtain datasets with multiple sizes. PDD [9] generates multiple small batches of synthetic images, each batch is conditioned on the accumulated data from previous batches. Unlike PDD, our current synthetic batch is independent with different operation iterations and not relevant to any previous batches. D3 [19] partitions large datasets into smaller subtasks and employs locally trained experts to distill each subset into distributions. These distilled distributions from the subtasks demonstrate effective generalization when applied to the entire task.

**Initialization.** Weight initialization [20, 21, 22, 23] is pivotal in training neural networks, significantly influencing their optimization process. Proper initialization is essential for ensuring model convergence and mitigating issues such as gradient vanishing. Recently, weight selection [24] introduces a strategy for initializing smaller models by selecting a subset of weights from a pretrained larger model. This

---

[3]Our synthetic images on ImageNet-1K are available anonymously at link.

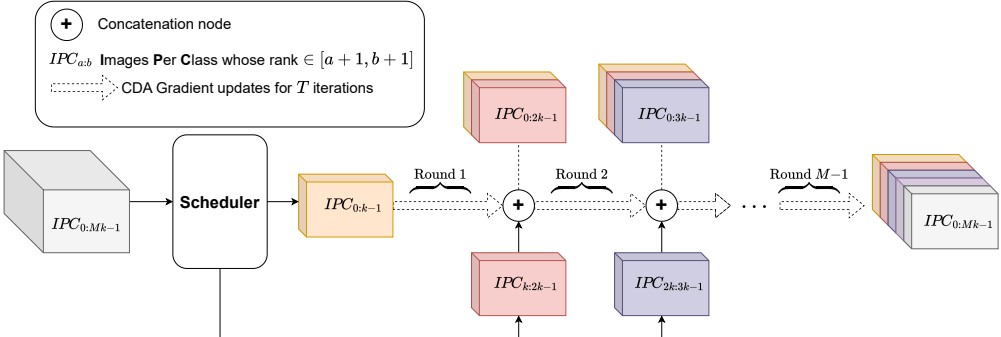

Figure 4: The proposed DELT learning procedure via a multi-round `EarlyLate` scheme.

method facilitates the transfer of learned attributes from the pretrained weights, enhancing the smaller model's performance. Weight subcloning [25] involves manipulating the pretrained model to derive a correspondingly scaled-down version with equivalent initialization. This involves two main steps: initially, it applies a neuron importance ranking to reduce the embedding dimension per layer within the pretrained model. Subsequently, it eliminates blocks from the transformer model to align with the layer count of the scaled-down network.

This work focuses on data initialization for generation processes. Few studies have examined this angle. While, PCA-K [26] appears to be the most relevant. It employs an initialization method that involves drawing samples from a distribution that accurately mirrors and is easily sampled from the training distribution. During training, it is possible to retrieve some details from the original image using the initial noisy sample, which at best provides a blurred representation of the original image.

## 3 Our Approach

**Preliminaries.** The objective of a regular dataset distillation task is to generate a compact synthetic dataset $\mathcal{S} = \{(\hat{\boldsymbol{x}}_1, \hat{\boldsymbol{y}}_1), \ldots, (\hat{\boldsymbol{x}}_{|\mathcal{S}|}, \hat{\boldsymbol{y}}_{|\mathcal{S}|})\}$ as a *student* dataset that captures a substantial amount of the information from a larger labeled dataset $\mathcal{T} = \{(\boldsymbol{x}_1, \boldsymbol{y}_1), \ldots, (\boldsymbol{x}_{|\mathcal{T}|}, \boldsymbol{y}_{|\mathcal{T}|})\}$, which serves as the *teacher* dataset. Here, $\hat{\boldsymbol{y}}$ represents the soft label for the synthetic sample $\hat{\boldsymbol{x}}$, and the size of $\mathcal{S}$ is much smaller than $\mathcal{T}$, yet it retains the essential information of the original dataset $\mathcal{T}$. The learning goal using this distilled dataset is to train a post-validation model with parameters $\boldsymbol{\theta}$:

$$\boldsymbol{\theta}_\mathcal{S} = \arg\min_{\boldsymbol{\theta}} \mathcal{L}_\mathcal{S}(\boldsymbol{\theta}), \tag{4}$$

$$\mathcal{L}_\mathcal{S}(\boldsymbol{\theta}) = \mathbb{E}_{(\hat{\boldsymbol{x}}, \hat{\boldsymbol{y}}) \in \mathcal{S}} \left[ \ell(\phi_{\boldsymbol{\theta}_\mathcal{S}}(\hat{\boldsymbol{x}}), \hat{\boldsymbol{y}}; \boldsymbol{\theta}) \right], \tag{5}$$

where $\ell$ is a standard loss function such as soft cross-entropy and $\phi_{\boldsymbol{\theta}_\mathcal{S}}$ represents the model.

The primary aim of dataset distillation is to produce synthetic data that ensures minimal performance difference between models trained on the synthetic dataset $\mathcal{S}$ and those trained on the original dataset $\mathcal{T}$ using validation data $V$. The optimization procedure for generating $\mathcal{S}$ is given by:

$$\arg\min_{\mathcal{S}, |\mathcal{S}|} \left( \sup \left\{ \left| \ell\left(\phi_{\boldsymbol{\theta}_\mathcal{T}}(\boldsymbol{x}_{val}), \boldsymbol{y}_{val}\right) - \ell\left(\phi_{\boldsymbol{\theta}_\mathcal{S}}(\boldsymbol{x}_{val}), \boldsymbol{y}_{val}\right) \right| \right\}_{(\boldsymbol{x}_{val}, \boldsymbol{y}_{val}) \sim V} \right). \tag{6}$$

where $(\boldsymbol{x}_{val}, \boldsymbol{y}_{val})$ are the sample and label pairs in the validation set of the real dataset $\mathcal{T}$. The learning task then focuses on the <data, label> pairs within $\mathcal{S}$, maintaining a balanced representation of distilled data across each class.

**Initialization.** Previous dataset distillation methods [11, 14, 12] on large-scale datasets like ImageNet-1K and 21K employ Gaussian noise by default for data initialization in the synthesis phase. However, Gaussian noise is random and lacks any semantic information. Intuitively, using real images provide a more meaningful and structured starting point, and this structured start can lead to quicker convergence during optimization because the initial data already contains useful features and patterns that are closer to the target distribution, which further

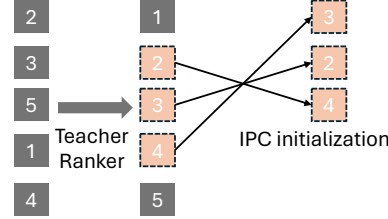

Figure 5: Selection criteria with a teach ranker.

enhances realism, quality, and generalization of the synthesized images. As shown in Fig. 2 right subfigure, our generated images exhibit both diversity and a high degree of realism in some cases.

**Selection Criteria.** Here, we introduce how to select real image patches to initialize the synthetic images. In our final syntheses, a significant fraction of our data has been subject to limited optimization iterations, making effective initialization crucial. A proper initialization also dramatically minimizes the overall computational load required for the updating on data. Prior approach [15] has demonstrated that choosing representative data patches from the original dataset without training can yield favorable performance without any additional training. Our observation, however, underscores that applying iterative refinement to original patches can lead to markedly improved results. As illustrated in Fig. 1, our selection criterion is based on a pretrained teacher model as a ranker, we calculate all patches' probabilities and sort them as the initialization pool. Then, we choose lowest, medium, or highest probability patches as the initialization for our optimization.

**Diversity-driven IPC Concatenation Training.** As shown in Fig. 4, to further emphasize diversity and avoid potential distribution bias from initialization, we optimize the initialized images starting from different points. The motivation behind this design is that different data samples require varying numbers of iterations to converge which is similar to the early stopping idea [27]. Importantly, as images become easier to predict with more updates by class labels, training primarily on easy data points can hinder model generalization. Therefore, our method enhances generalization by generating data samples with varying difficulty levels, acting as a regularizer by limiting the optimization process to a smaller volume of image pixel space. Previous work [28] studies how to perform early stopping training on different layers' weights of the model with progressive retraining to mitigate noisy labels. We are pioneering to study how to leverage early-late training when optimizing data. Moreover, we improve the efficiency of our approach by performing gradient updates in a single scan. Initially, we conduct a single gradient loop, continually introducing new data for distillation by concatenating them at different time stamps. Consequently, the $M$ batch receives the synthetic images of all preceding batches, $IPC_{0:Mk-1}$, as final generations. This process can be simplified as follows:

$$IPC_{0:Mk-1} = [\underbrace{\hat{\boldsymbol{x}}_0, \hat{\boldsymbol{x}}_1, \ldots, \hat{\boldsymbol{x}}_{k-1}}_{IPC_{0:k-1}}, \ldots, \hat{\boldsymbol{x}}_{Mk-1}] \qquad (7)$$

$$\underbrace{\hspace{5cm}}_{IPC_{0:Mk-1}}$$

where $[\hat{\boldsymbol{x}}_0, \hat{\boldsymbol{x}}_1, \ldots, \hat{\boldsymbol{x}}_{Mk-1}]$ refers to the concatenation of the generated image. $M$ is the number of batches, $k$ is the number of generated images in each batch. We train these different batches at different starting points, each batch goes through a completed learning phase, but the total number of iterations varies. Then, the multiple IPCs of $\hat{\boldsymbol{x}}$ are concatenated into a simple batch. Because of its early-late training property, we refer to this simple training scheme as `EarlyLate` training.

**Training Procedure.** As illustrated in Fig. 4, our learning procedure is extremely simple using an incremental learning process: We split the total IPCs to be learned into multiple batches. The training begins with the first batch. Following a predefined number of iterations, the second batch commences its iterative training, and this process continues sequentially with subsequent batches. *Batch-to-global* matching algorithm [12] of Eq. 3 has been utilized between each round.

## 4 Experiments

### 4.1 Datasets and Results Details

We first run DELT on five standard benchmark tests including CIFAR-10 (10 classes) [29], Tiny-ImageNet (200 classes) [30], ImageNet-1K (1,000 classes) [31] and it variants of ImageNette (10 classes) [32], and ImageNet-100 (100 classes) [33] with performances reported in Table 1 and Table 2. The evaluation protocol is following prior works [15, 11]. We compare DELT to six baseline dataset distillation algorithms including Matching Training Trajectories (MTT) [4], Improved Distribution Matching (IDM) [34], TrajEctory Matching with Constant Memory (TESLA) [8], Squeeze-Recover-Relabel (SRe$^2$L) [11], Difficulty-Aligned Trajectory-Matching (DATM) [35], Realistic-Diverse-Efficient Dataset Distillation (RDED) [15]. Following previous dataset distillation methods [2, 15, 11], we use ConvNet [36], ResNet-18/ResNet-101 [37], EfficientNet-B0 [38], MobileNet-V2 [39], MnasNet1_3 [40], and RegNet-Y-8GF [41], as our backbone for training or post-validation. All our experiments are conducted on 4 NVIDIA RTX 4090 GPUs.

| Dataset | IPC | ResNet-18 | | | ResNet-101 | | | MobileNet-v2 |
|---|---|---|---|---|---|---|---|---|
| | | SRe$^2$L [11] | RDED [15] | Ours | SRe$^2$L [11] | RDED [15] | Ours | Ours |
| CIFAR-10 | 1 | 16.6 ± 0.9 | 22.9 ± 0.4 | **24.0 ± 0.8** | 13.7 ± 0.2 | 18.7 ± 0.1 | **20.4 ± 1.0** | 20.2 ± 0.4 |
| | 10 | 29.3 ± 0.5 | 37.1 ± 0.3 | **43.0 ± 0.9** | 24.3 ± 0.6 | 33.7 ± 0.3 | **37.4 ± 1.2** | 29.3 ± 0.3 |
| | 50 | 45.0 ± 0.7 | 62.1 ± 0.1 | **64.9 ± 0.9** | 34.9 ± 0.1 | 51.6 ± 0.4 | **54.1 ± 0.8** | 42.9 ± 2.2 |
| ImageNette | 1 | 19.1 ± 1.1 | **35.8 ± 1.0** | 24.1 ± 1.8 | 15.8 ± 0.6 | **25.1 ± 2.7** | 19.4 ± 1.7 | **19.1 ± 1.0** |
| | 10 | 29.4 ± 3.0 | 61.4 ± 0.4 | **66.0 ± 1.4** | 23.4 ± 0.8 | 54.0 ± 0.4 | 55.4 ± 6.2 | 64.7 ± 1.4 |
| | 50 | 40.9 ± 0.3 | 80.4 ± 0.4 | **88.2 ± 1.2** | 36.5 ± 0.7 | 75.0 ± 1.2 | 83.3 ± 1.1 | 85.7 ± 0.4 |
| Tiny-ImageNet | 1 | 2.62 ± 0.1 | **9.7 ± 0.4** | 9.3 ± 0.5 | 1.9 ± 0.1 | 3.8 ± 0.1 | **5.6 ± 1.0** | **3.5 ± 0.5** |
| | 10 | 16.1 ± 0.2 | 41.9 ± 0.2 | **43.0 ± 0.1** | 14.6 ± 1.1 | 22.9 ± 3.3 | **42.8 ± 0.9** | 26.5 ± 0.5 |
| | 50 | 41.1 ± 0.4 | **58.2 ± 0.1** | 55.7 ± 0.5 | 42.5 ± 0.2 | 41.2 ± 0.4 | **58.5 ± 0.3** | 51.3 ± 0.5 |
| ImageNet-100 | 10 | 9.5 ± 0.4 | **36.0 ± 0.3** | 28.2 ± 1.5 | 6.4 ± 0.1 | **33.9 ± 0.1** | 22.4 ± 3.3 | 15.8 ± 0.2 |
| | 50 | 27.0 ± 0.4 | 61.6 ± 0.1 | **67.9 ± 0.6** | 25.7 ± 0.3 | 66.0 ± 0.6 | **70.8 ± 2.3** | 55.0 ± 1.8 |
| | 100 | - | 74.5 ± 0.4 | **75.1 ± 0.2** | - | 73.5 ± 0.8 | **77.6 ± 1.8** | 76.7 ± 0.3 |
| ImageNet-1K | 10 | 21.3 ± 0.6 | 42.0 ± 0.1 | **45.8 ± 0.1** | 30.9 ± 0.1 | 48.3 ± 1.0 | **48.5 ± 1.6** | 35.1 ± 0.5 |
| | 50 | 46.8 ± 0.2 | 56.5 ± 0.1 | **59.2 ± 0.4** | 60.8 ± 0.5 | 61.2 ± 0.4 | **66.1 ± 0.5** | 56.2 ± 0.3 |
| | 100 | 52.8 ± 0.3 | 59.8 ± 0.1 | **62.4 ± 0.2** | 62.8 ± 0.2 | - | **67.6 ± 0.3** | 58.9 ± 0.3 |

Table 1: Comparison with SOTA dataset distillation methods using relatively large-scale backbones on five benchmarks across different scales. MobileNet-v2 is modified to match the low resolutions of CIFAR-10 and Tiny-ImageNet following [42]. Due to the table space limitation, some other methods that are weaker than RDED are not listed, such as CDA and G-VBSM. Since IPC 1 is not applicable to use `EarlyLate` strategy and the single image in each class is optimized with a constant iteration.

| Dataset | IPC | ConvNet | | | | | |
|---|---|---|---|---|---|---|---|
| | | MTT [4] | IDM [34] | TESLA [8] | DATM [35] | RDED [15] | Ours |
| ImageNette | 1 | **47.7 ± 0.9** | - | - | - | 33.8 ± 0.8 | 29.8 ± 1.4 |
| | 10 | 63.0 ± 1.3 | - | - | - | **63.2 ± 0.7** | 51.7 ± 1.2 |
| | 50 | - | - | - | - | 83.8 ± 0.2 | **84.5 ± 0.4** |
| Tiny-ImageNet | 1 | 8.8 ± 0.3 | 10.1 ± 0.2 | - | **17.1 ± 0.3** | 12.0 ± 0.1 | 12.4 ± 0.8 |
| | 10 | 23.2 ± 0.2 | 21.9 ± 0.3 | - | 31.1 ± 0.3 | 39.6 ± 0.1 | **40.0 ± 0.4** |
| | 50 | 28.0 ± 0.3 | 27.7 ± 0.3 | - | 39.7 ± 0.3 | 47.6 ± 0.2 | **48.6 ± 0.2** |
| ImageNet-100 | 10 | - | 17.1 ± 0.6 | - | - | **29.6 ± 0.1** | 24.7 ± 1.5 |
| | 50 | - | 26.3 ± 0.4 | - | - | 50.2 ± 0.2 | **51.9 ± 1.1** |
| | 100 | - | - | - | - | 58.6 ± 0.4 | **61.5 ± 0.5** |
| ImageNet-1K | 1 | - | - | 7.7 ± 0.2 | - | 6.4 ± 0.1 | **8.8 ± 0.5** |
| | 10 | - | - | 17.8 ± 1.3 | - | 20.4 ± 0.1 | **31.3 ± 0.8** |
| | 50 | - | - | 27.9 ± 1.2 | - | 38.4 ± 0.2 | **41.7 ± 0.1** |

Table 2: Comparison with SOTA dataset distillation methods using small-scale backbone architecture on four benchmark datasets. Following [4, 34, 15], Conv-3 is used for CIFAR-10, Conv-4 for Tiny-ImageNet and ImageNet-1K, Conv-5 for ImageNette, and Conv-6 for ImageNet-100 and ImageNet-1K. Entries marked with "-" are missing due to scalability issue.

As shown in Table 1, our approach establishes the new state-of-the-art accuracy in 13 out of 15 of the configurations on five datasets from small-scale CIFAR-10 to large-scale ImageNet-1K using relatively large backbone architecture of ResNet-101, in many cases with significant margins of improvement. The results using small-scale architecture ConvNet are shown in Table 2, our approach also achieves the state-of-the-art accuracy in 8 out of 12 of the configurations on four datasets.

### 4.2 Cross-architecture generalization

An important characteristic of distilled datasets is their effectiveness in generalizing to novel training architectures. In this context, we assess the transferability of DELT's distilled datasets tailored for ImageNet-1K with 10 images per class. Following previous studies [11, 15], we test our models using five distinct architectures: ResNet-18 [37], MobileNet-V2 [39], MnasNet1_3 [40], EfficientNet-B0 [38], and RegNet-Y-8GF [41]. As shown in Table 4, our proposed approach demonstrates significant better performance than other competitive methods on all these architectures.

### 4.3 Ablation Study

**Mosaic splicing pattern.** Mosaic stitching method [43] in RDED selects four crops from the train set as the optimal hyper-parameter, and puts the contents of the four crops into a synthetic image that is directly used for post-validation. In this work, considering that we use different difficulty levels of

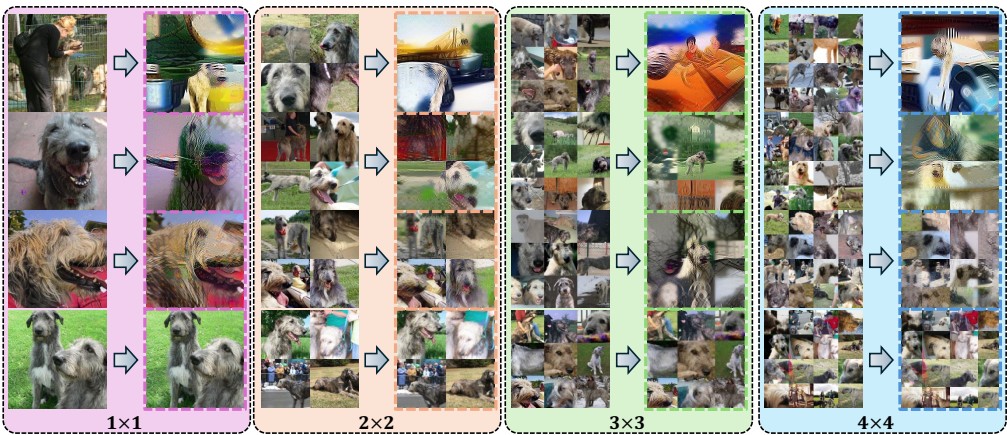

Figure 6: Mosaic splicing patterns on ImageNet-1K using real image patches as the initialization. In each block, the left column is the starting real image initialized samples and right is the final optimized syntheses. From top to bottom are images generated by early training and late training.

selection for initialization, we examine different strategies of the Mosaic splicing patterns, including $1 \times 1$, $2 \times 2$, $3 \times 3$, $4 \times 4$, and $5 \times 5$ patches, as illustrated in Fig. 11. The ablation results are shown in Table 3, it can be observed that $1 \times 1$ achieves the best accuracy.

**Initialization.** We examine how different initialization strategies affect final performance, including: choosing lowest probability crops, medium probability crops and highest probability crops. Our results are shown in Table 3. Overall, the performance gap between different strategies is not significant, and selecting the medium probability crops as the initialization achieves the best accuracy.

**Optimization iterations.** We examine two types of optimization iterations: maximum iteration (MI) for the earliest batch training and round iteration (RI). MI presents the number of optimization iterations that the earliest batch goes through. RI represents the number of iterations used for each round in Fig. 4. It essentially indicates the iteration gap between the optimization of two adjacent batches. As shown in Table 3, we test MI values of 1K, 2K, and 4K, using 500 and 1K iterations for each RI. Note that when MI is set to 1K, it is not feasible to use 1K as RI. The results show that 4K (same as [11, 12]) MI and 500 RI achieves the best accuracy.

**Early-only vs. EarlyLate.** Early-only is equivalent to using constant MI to optimize each image. The method will transform to baseline *batch-to-global* matching of CDA [12] + real image initialization. Our results in Table 3 clearly show that the `EarlyLate` training bring a significant improvement on final performance. More importantly, this strategy is the key factor in enhancing generation diversity.

**Real image stitching vs. Minimax diffusion vs. Ours.** We further compare the performance of our approach with real image stitching [15] and diffusion generation [44]. The results are presented in Table 3d. While the first two methods produce more realistic images, each image contains limited information. In contrast, our method achieves the best final performance.

### 4.4 Computational Analysis

For image optimization-based methods like SRe$^2$L and CDA, the total computational cost is calculated as $N \times T$, where $N$ is the MI. In our `EarlyLate` scheme, the first batch images undergo $T_1$ iterations (where $T_1 = T$). Subsequent batches are processed with progressively fewer iterations, such as $T_2$ ($T_2 = T_1 - RI$) for the next set, and so forth. The iterations for the final batch are reduced to RI which is $1/j$ of the standard count (where $j = 4$ or 8 in our ablation), the total number of our optimization iterations required is $N \times T - \frac{j(j-1)}{2} RI$, which is roughly 2/3 of prior *batch-to-global* matching methods. Our real time consumptions for data generation are shown in Table 5, note that the smaller the dataset like CIFAR, the more time is spent on loading and processing the data, rather than training.

### 4.5 Visualization of DELT

Fig. 7 illustrates a comprehensive visual comparison between randomly selected synthetic images from our distilled dataset and those from the real image patches [15], MinimaxDiffusion [44], MTT [4], IDC [45], SRe$^2$L [11], SCDD [46], CDA [12] and G-VBSM [14] distilled data. It can be observed that

Table 3: **Ablation experiments** on various aspects of our framework with ResNet-18 on ImageNet-1K.

| # Patches | Top 1 acc |
|-----------|-----------|
| $1 \times 1$ | **57.57** |
| $2 \times 2$ | 56.92 |
| $3 \times 3$ | 56.62 |
| $4 \times 4$ | 56.71 |
| $5 \times 5$ | 56.51 |

(a) **Number of patches**. Ablation on initializing different numbers of scoring patches. Results are from ResNet-18 on ImageNet-1K for 500 iterations to synthesize 50 IPCs.

| Selection criteria | Top 1 acc |
|--------------------|-----------|
| Lowest probability | 57.55 |
| Medium probability | **57.67** |
| Highest probability | 57.03 |

(b) **Selection criteria**. Initializing $1 \times 1$ images selected according to teacher model's probability

| Iterations | Round Iterations 500 | 1K |
|------------|-----|-----|
| 1K | 44.87 | n/a |
| 2K | 45.61 | 44.40 |
| 4K | **46.42** | 44.66 |

(c) **Round Iterations**. Top-1 acc. of our method for IPC 10 using different round iterations with ResNet-18.

| Dataset | CDA [12] + Our init. | Ours |
|---------|----------------------|------|
| ImageNet-1K | 43.5 | **45.8** |
| Tiny-ImageNet | 42.2 | **43.0** |
| CIFAR-10 | 39.4 | **43.0** |

(d) Ablation on init. and `EarlyLate` under IPC 10.

| IPC | RDED [15] | MinimaxDiffusion [44] | Ours |
|-----|-----------|-----------------------|------|
| 10 | 42.0 | 44.3 | **45.8** |
| 50 | 56.5 | 58.6 | **59.2** |

(e) Comparison with real and diffusion generated data.

Table 4: **Cross-architecture generalization**. Results are evaluated on IPC 10.

| Recover \Validation | | ResNet-18 | EfficientNet-B0 | MobileNet-V2 | MnasNet1_3 | RegNet-Y-8GF |
|---------------------|------|-----------|------------------|--------------|------------|--------------|
| ResNet-18 | SRe$^2$L [11] | 41.9 | 41.9 | 33.1 | 39.3 | 51.5 |
| | CDA [12] | 42.2 | 43.9 | 34.2 | 39.7 | 52.9 |
| | G-VBSM [14] | 41.4 | 42.6 | 33.5 | 40.1 | 52.2 |
| | RDED [15] | 42.3 | 42.8 | 34.4 | 40.0 | 54.8 |
| | Ours | **46.4**$_{(+4.1)}$ | **47.1**$_{(+4.3)}$ | **36.1**$_{(+1.7)}$ | **40.7**$_{(+0.7)}$ | **57.5**$_{(+2.7)}$ |

Table 5: **Actual computational consumption and analysis** (hours under IPC 50) in data synthesis with image optimization-based methods on a single NVIDIA 4090 GPU. "RI" represents *round iterations*. A total 4K iterations are used for all methods and datasets to ensure fair comparisons.

| | Dataset (hours) | | |
|--------|-----------------|---------------|-----------|
| Method | ImageNet-1K | Tiny-ImageNet | CIFAR-10 |
| G-VBSM [14] | 114.1 | 5.5 | 0.195 |
| SRe$^2$L [11] | 29.0 | 5.0 | 0.084 |
| CDA [12] | 29.0 | 5.0 | 0.084 |
| Ours (RI = 500) | **17.6**$_{(\downarrow 39.3\%)}$ | **3.4**$_{(\downarrow 32.0\%)}$ | **0.083**$_{(\downarrow 1.1\%)}$ |
| Ours (RI = 1K) | **18.8**$_{(\downarrow 35.2\%)}$ | **3.6**$_{(\downarrow 28.0\%)}$ | **0.084**$_{(\downarrow 0.0\%)}$ |

the images generated by each method have their own characteristics. MinimaxDiffusion leverages the diffusion model to synthesize images which is close to the real ones. However, as in our above ablation, both real and diffusion-generated data are inferior to ours. MTT results show noticeable artifacts and distortions, the objects in all images are located in the middle of the generations, the diversity is limited. IDC results also show distorted and less recognizable dog images, but diversity is increased. SRe$^2$L exhibits some dog features but with significant distortions and similar simple background. SCDD shows more recognizable dog features but still the color is simple and monochromatic, the same situation happens in CDA. G-VBSM shows more colorful patterns, possibly due to recovery from multiple different networks, but all generations are in the same pattern and the diversity is not large. Our approach's synthetic images exhibit a higher degree of diversity, including both compressed distorted images from long-optimized initializations and clear, recognizable dog images from short-optimized initializations, a unique capability not present in other methods.

## 4.6 Application I: Data-free Network Pruning

Our distilled dataset acts as a multifunctional training tool and boosts the adaptability for diverse downstream applications. We validate its utility in the scenario of data-free network pruning [47]. Table 6 shows the applicability of our dataset in this task when pruning 50% weights, where it significantly surpasses previous methods such as SRe$^2$L and RDED under IPC 10 and 50.

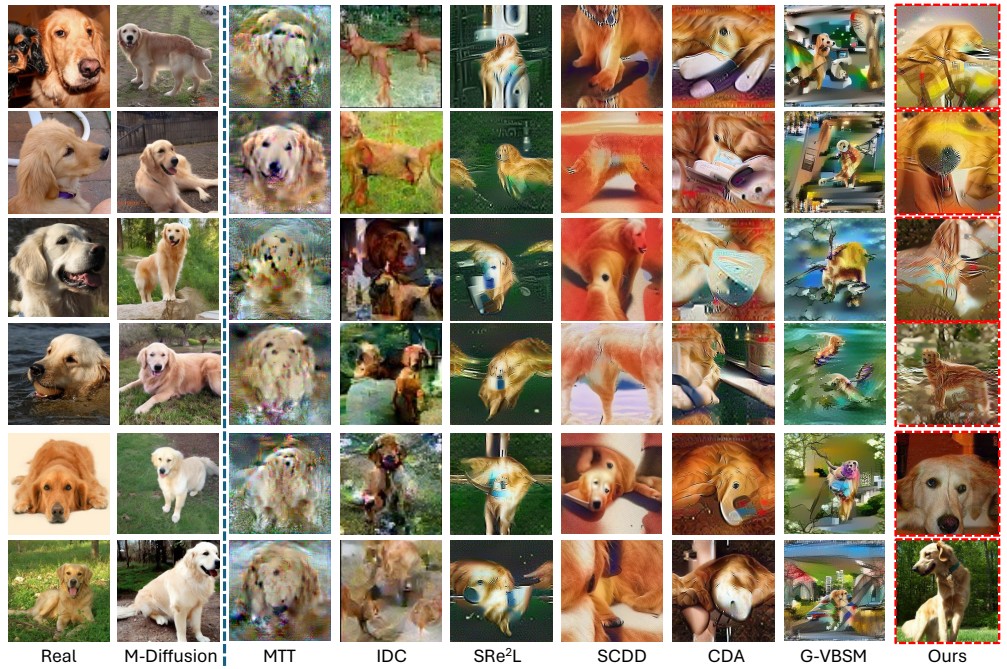

| Real | M-Diffusion | MTT | IDC | SRe²L | SCDD | CDA | G-VBSM | Ours |

Figure 7: Distilled dataset visualization compared with other image optimization-based methods.

Table 6: Accuracy of data-free network pruning using slimming [48] on VGG11-BN [49].

|        | SRe²L [11] | RDED [15] | Ours          |
|--------|------------|-----------|---------------|
| IPC 10 | 12.5       | 13.2      | **17.9**(+4.7) |
| IPC 50 | 31.7       | 42.8      | **44.8**(+2.0) |

## 4.7 Application II: Continual Learning

We examine the effectiveness of DELT generated images in the continual learning scenario. Following the setup in prior studies [11, 6], we perform 100-step class-incremental experiments on ImageNet-1K, comparing our results with the baselines G-VBSM and SRe²L. As shown in Fig. 8, our DELT distilled dataset significantly outperforms G-VBSM, with an average improvement of about 10% in 100-step class-incremental learning task. This highlights the significant benefits of deploying DELT, particularly in mitigating the challenges of continual learning.

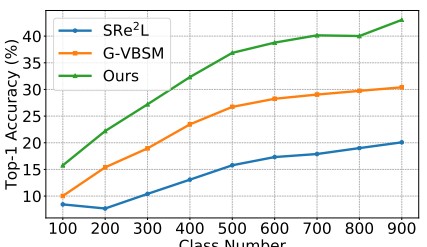

Figure 8: Continual learning results.

## 5 Conclusion

We have introduced a new training strategy, `EarlyLate`, to improve image diversity in *batch-to-global* matching scenarios for dataset distillation. The proposed approach organizes predefined IPC samples into smaller, manageable subtasks and utilizes local optimizations. This strategy helps in refining each subset into distributions characteristic of different phases, thereby mitigating the homogeneity typically caused by a singular optimization process. The images refined through this method exhibit robust generalization across the entire task. We have extensively evaluated this approach on CIFAR-10 and 100, Tiny-ImageNet, ImageNet-1K, and its variants. Our empirical findings indicate that our approach significantly outperforms prior state-of-the-art methods across various IPC configurations.

**Limitations.** Our method effectively avoids the issue of insufficient data diversity generated by *batch-to-global* methods and reduces the computational cost of the generation process. However, there is still a performance gap when training the model on our generated data compared to training on the original dataset. Additionally, our short-optimized data exhibits similar semantic information to the original images, which may potentially leak the privacy of the original dataset.

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

# Appendix

## A   Broader Impacts

Our dataset distillation framework can significantly reduce the computational resources required for training machine learning models. This leads to lower energy consumption and cost, making AI more accessible and sustainable. By generating smaller, more manageable datasets, researchers and developers can iterate and experiment more quickly, accelerating the pace of innovation in various AI applications. However, condensed datasets might inadvertently amplify biases present in the original data. If the distillation process does not adequately address bias, it could lead to unfair or discriminatory AI systems. Also, simplifying datasets may lead to a loss of important nuances and context, potentially degrading the performance of models in real-world applications where such details are crucial. Moreover, the models may overfit to condensed data, indicating that models trained on distilled datasets might perform well on the condensed data but poorly on more diverse real-world data, limiting their generalizability and robustness.

## B   Training Details

Table 7: Hyper-parameter settings.

(a) Validation settings

| config | value |
| --- | --- |
| optimizer | AdamW |
| base learning rate | 0.001 (all) |
| | 0.0025 (MobileNet-v2) |
| weight decay | 0.01 |
| | 100 (IPC50) |
| batch size | 50 (IPC10) |
| | 10 (IPC1) |
| learning rate schedule | cosine decay |
| training epoch | 300 |
| | RandAugment |
| augmentation | RandomResizedCrop |
| | RandomHorizontalFlip |

(b) Recovery settings

| config | value |
| --- | --- |
| $\alpha_{\text{BN}}$ | 0.01 |
| optimizer | Adam |
| base learning rate | 0.25 |
| momentum | $\beta_1, \beta_2 = 0.5, 0.9$ |
| batch size | 100 |
| learning rate schedule | cosine decay |
| recovery iteration | 4,000 |
| round iteration | 500 [IPC 10, 50, 100] |
| initialization | top medium |
| augmentation | RandomResizedCrop |

(c) Dataset-specific settings in recovery

| config | CIFAR10 | Tiny-ImageNet | ImageNette | ImageNet-100 | ImageNet-1K |
| --- | --- | --- | --- | --- | --- |
| RandAugment (m) | 5 | 4 | 6 | 6 | 6 |
| RandAugment (n) | 4 | 3 | 2 | 2 | 2 |
| RandAugment (mstd) | 1.0 | 1.0 | 1.0 | 1.0 | 1.0 |
| IPC1 Recovery Iterations | 2K (R18) | 500 (R18) | 1K (R18) | - | 3K (Conv4) |
| | 3K (R101) | 500 (R101) | 1K (R101) | - | - |
| | 2K (MobileNet) | 500 (MobileNet) | 2K (MobileNet) | - | - |
| | - | 1K (Conv4) | 4K (Conv5) | - | - |

For reproducibility, we provide all our hyper-parameter settings used in our experiments in Table 7, we outline such details below.

**Squeezing and Pre-trained models.** Following the previous works [11, 12, 15], we use the official PyTorch [50] pre-trained ResNet-18 model for ImageNet-1K, and we use the same official Torchvision [50] code to produce our pre-trained models, ResNet-18 and ConvNet, for the other datasets.

**Ranking.** A crucial part of our method is initialization, we simply use ResNet-18 pre-trained models to rank and select the top-medium images as initialization for all our datasets, except for ImageNet-100 where we simply extracted the top-medium images based on the rankings of the original ImageNet-1K.

**Recovery.** For our synthesis, we provide the details of the general hyper-parameters used for different datasets, including ImageNet-1K, ImageNet-100, ImageNette, Tiny-ImageNet, and CIFAR10, in Table 7b. Because synthesizing a single image per class, i.e., IPC 1, is quite special as we cannot use

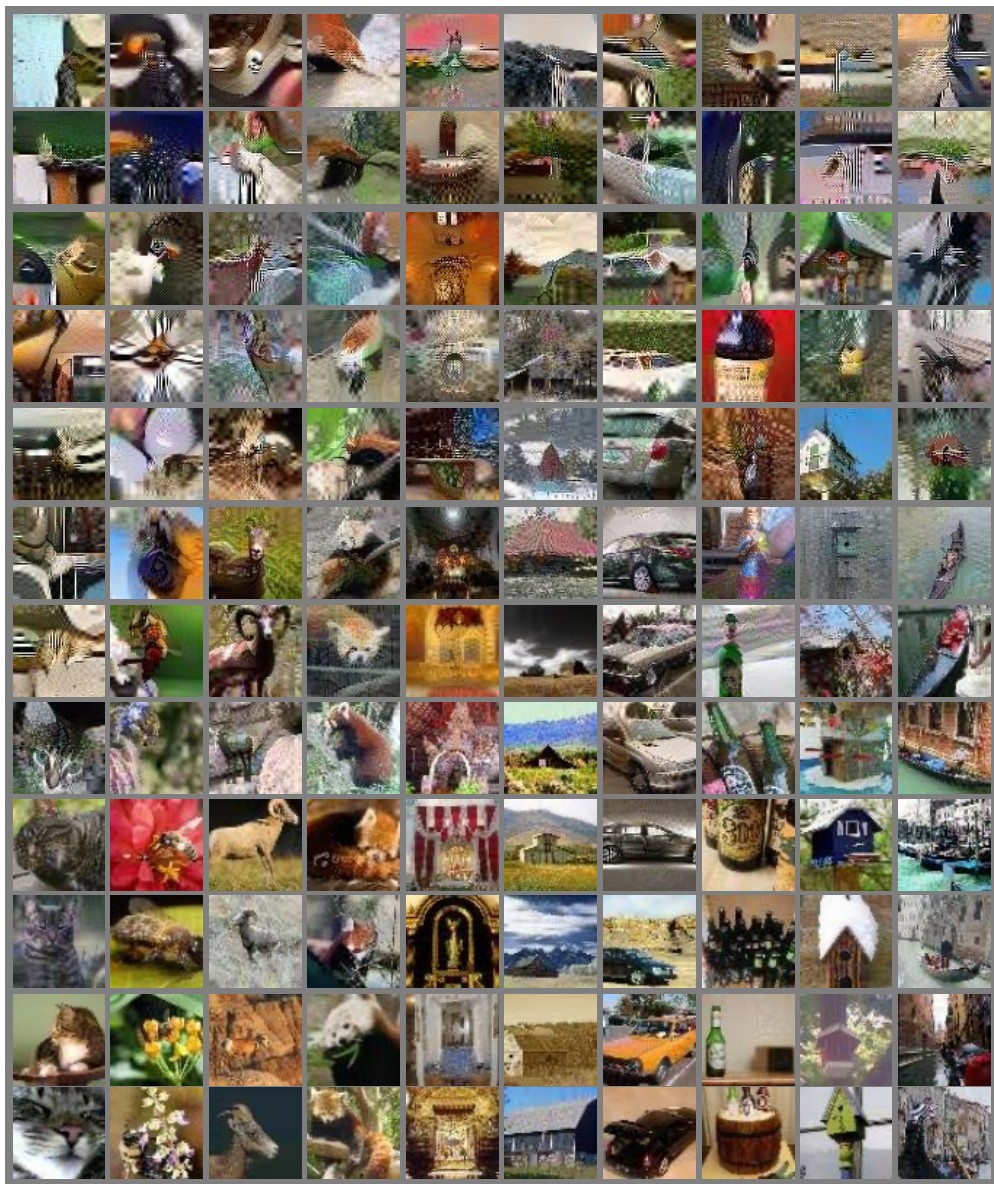

Figure 9: Synthetic image visualizations on Tiny-ImageNet generated by our DELT.

rounds, we apply different numbers of iterations based on both the dataset scale and the validation teacher model as outlined in Table 7c.

**Validation.** This includes both the soft-label generation, Relabel in SRe$^2$L, and evaluation, or post-training. We outline such details in Table 7a. We use `timm`'s version of RandAugment [51] with different settings depending on the synthesized dataset being validated as outlined in Table 7c.

## C   More Visualizations

We provide more visualizations on synthetic Tiny-ImageNet, ImageNette and CIFAR-10 datasets. In each figure, each column represents a different class, with images progressing from long optimization at the top to short optimization at the bottom.

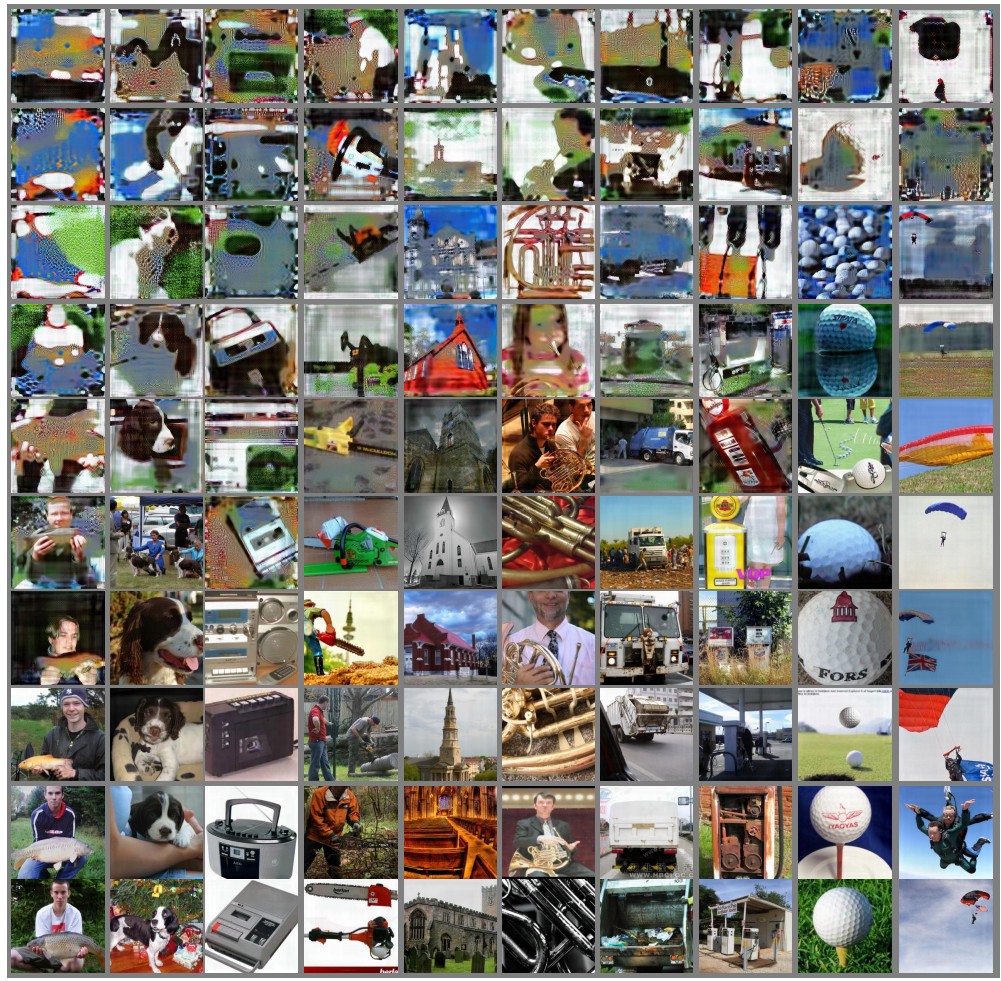

Figure 10: Synthetic image visualizations on ImageNette generated by our DELT.

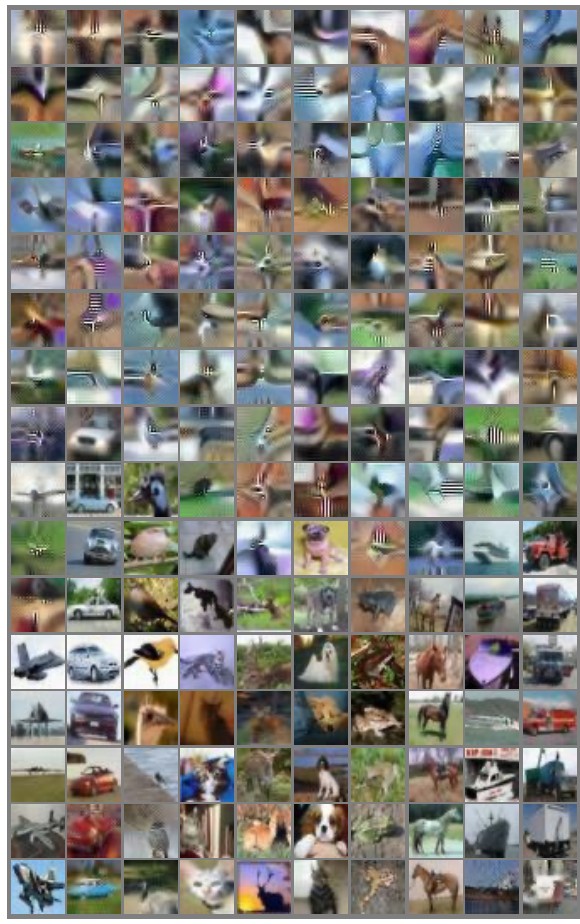

Figure 11: Synthetic image visualizations on CIFAR-10 generated by our DELT.

