# OpenReview forum: "DELT: A Simple Diversity-driven EarlyLate Training for Dataset Distillation"
_NeurIPS.cc/2024/Conference — Submitted to NeurIPS 2024_

### Official Review · Reviewer_1mgP · 2024-07-09

**Soundness:** 3
**Presentation:** 3
**Contribution:** 2
**Rating:** 5
**Confidence:** 4

**Summary:**

The paper presents a simple but novel approach to enhance image diversity in dataset distillation. previous methods face challenges in balancing computational efficiency and diversity in synthetic images. The proposed EarlyLate training scheme addresses these issues by partitioning predefined IPC samples into smaller subtasks and using local optimizations for each subset.

**Strengths:**

1. The EarlyLate training scheme effectively enhances the diversity of synthetic images. This is a very simple but novel approach to increase the diversity of distilled datasets. I believe it can provide some inspiration for future work.
2. The method, or the training scheme reduces the computational load compared to batch-to-global matching methods.
3. The experiments are comprehensive, including performance, cross-architecture generalization, ablation, and application. These experiments verify the method's superiority.

**Weaknesses:**

1. Compared to previous methods, the work in this paper is incremental.
2. The motivation and the advantages of the "Selection Criteria" in the initialization approach are not clear. And I am confused about how to rank, which is presented in Fig 5, could the authors explain it here?
3. There are a lot of hyperparameters involved. How should these hyperparameters be tuned? Are there any principled approaches?
4. I want to know the impact of the initialization method. In the ablation study, only CDA+init is shown. More advanced methods with init and whether EarlyLate uses init are not presented.
5. The performance of other sota methods on MobileNet-v2 is not presented in Table 1. Is the proposed method still better than other sota methods on MobileNet-v2?
6. Training tricks like random crop play a significant role in methods such as SRe2L. I would like to know to what extent the method proposed in this paper relies on such tricks.

**Questions:**

Please refer to weaknesses.

**Limitations:**

The authors have adequately addressed the limitations and potential negative societal impact of their work.

---

> ### Author Rebuttal · Authors · 2024-08-07
>
> Thank you for the valuable and constructive comments. We will incorporate all suggestions in our revision. Below, we provide further clarifications to the reviewer's questions.
>
> >Q1: Comparison to previous methods.
>
> Thanks for your comments. We are confident that no prior work has specifically focused on *EarlyLate* training in the context of dataset distillation. While our method is technically straightforward, it achieves state-of-the-art accuracy across multiple benchmarks, from small-scale to large-scale datasets. Therefore, we believe our contribution is significant and not merely incremental for dataset distilaltion task.
>
> >Q2: The motivation and the advantages of the "Selection Criteria" in the initialization approach are not clear. And I am confused about how to rank, which is presented in Fig 5, could the authors explain it here?
>
> Thanks for highlighting this point. We select the N images with scores around the median from the teacher model: the score being the probability of the true class. The motivation is that such images have medium difficulty level to the teacher, so they have more room for information improvement via distillation gradients. We further empirically validate this strategy by comparing different approaches in Table 3b.
>
> For instance, in Figure 5, ranking is based on the probability of the true class, and the figure shows selecting (IPC=3) images around the median score, to get the medium difficulty images.
>
> >Q3: There are a lot of hyperparameters involved. How should these hyperparameters be tuned? Are there any principled approaches?
>
> The original hyperparameters follow baseline methods of CDA and SRe$^2$L, our method have a hyper-parameter for the Round Iterations (RI) that specifies the number of rounds = Total Iterations / RI. As highlighted in Table 3.c, less round iterations generates more rounds and thus contributes to more diverse synthesized images and better performance.
>
> >Q4: I want to know the impact of the initialization method. In the ablation study, only CDA+init is shown. More advanced methods with init and whether EarlyLate uses init are not presented.
>
> Thanks for pointing this out, we hightlight that DELT, the EarlyLate, indeed uses the mentioned initialization. As for the other methods, we focused on gradient-based methods capable of scaling to large datasets. Therefore, we compared using the advanced CDA or SRe$^2$L, and the CDA was better. We provide the performance comparison below for IPC 50 on ImageNet-1K:
>
> | Initialization |  SRe$^2$L + w/ Init w/o EarlyLate | CDA + w/ Init w/o EarlyLate |  CDA + w/ Init + w/ EarlyLate (Our Method) |
> |:--------------:|:--------:|:--------:|:--------:|
> |       2x2      |  55.3     | 56.9       |  58.2 (**+1.3%**)  |
> |       3x3      |  55.8     | 56.6       |  58.1 (**+1.5%**) |
> |       4x4      |  55.2     | 56.7       |  57.4 (**+0.7%**) |
> |       5x5      |  54.6     | 56.5       |  57.3 (**+0.8%**) |
>
> As we can see, our *EarlyLate* strategy enhances the performance of around +1% over the initialization. Without initialization, our method improves even more, with 2.4% as follows:
>
> | Strategy |  SRe$^2$L w/o Init w/o EarlyLate |CDA w/o Init w/o EarlyLate |CDA w/o Init  w/ EarlyLate |
> |:-----------:|:--------:|:--------:|:--------:|
> |  | 46.8  |  53.5   |   55.9 (**+2.4%**)|
>
> In brief, initialization alone enhances the performance over the basic CDA/SRe$^2$L. The proposed *EarlyLate* strategy further enhances the performance by +1% over the initialization.
>
> [1] Squeeze, recover and relabel: Dataset condensation at imagenet scale from a new perspective. Advances in Neural Information Processing Systems, 2023.
> [2] Dataset Distillation in Large Data Era. arXiv preprint arXiv:2311.18838.
>
> >Q5: The performance of other sota methods on MobileNet-v2 is not presented in Table 1. Is the proposed method still better than other sota methods on MobileNet-v2?
>
> We appreciate your suggestion. We provide the comparison of our method against RDED when using MobileNet-v2 as below:
>
>  |    Dataset    | IPC |    RDED    |      DELT      |
> |:-------------:|:-------:|:----------:|:--------------:|
> |               |  1  | 18.1 ± 0.9 | **20.2 ± 0.4** |
> |    CIFAR10    |  10 | 29.2 ± 1.1 | **29.3 ± 0.3** |
> |               |  50 | 39.9 ± 0.5 | **42.9 ± 2.2** |
> |               |  1  | **26.4 ± 3.4** | 19.1 ± 1.0     |
> |   ImageNette  |  10 | 52.7 ± 6.6 | **64.7 ± 1.4** |
> |               |  50 | 80.0 ± 0.0 | **85.7 ± 0.4** |
> |               |  1  | **3.5 ± 0.1** | **3.5 ± 0.5**  |
> | Tiny-ImageNet |  10 | 24.6 ± 0.1 | **26.5 ± 0.5** |
> |               |  50 | 49.3 ± 0.2 | **51.3 ± 0.5** |
> |               |  50 | 51.5 ± 0.8 | **55.0 ± 1.8** |
> |  ImageNet-100 | 100 | 70.8 ± 1.1 | **76.7 ± 0.3** |
> |               |  10 | 32.3 ± 0.2 | **35.1 ± 0.5** |
> |  ImageNet-1K  |  50 | 52.8 ± 0.4 | **56.2 ± 0.3** |
> |               | 100 | 56.2 ± 0.1 | **58.9 ± 0.3** |
>
> >Q6: Training tricks like random crop play a significant role in methods such as SRe2L. I would like to know to what extent the method proposed in this paper relies on such tricks.
>
> Thank you for highlighting this point. We have included a table comparing different initial crop ranges in random crop augmentation for our DELT method. As shown in the results, the 0.08-1.0 range yields the best performance, which is why it is the default setting in our framework.
>
> |  Random Crop Range | Top 1-acc |
> |:------------------:|:---------:|
> |      0.08-1.0      |  **67.8** |
> |       0.2-1.0      |    67.3   |
> |       0.5-1.0      |    66.3   |
> |       0.8-1.0      |    66.3   |

---

> > ### Comment · Reviewer_1mgP · 2024-08-12
> >
> > Thanks for the reply. Some of my concerns are addressed and I will keep my score.

---

### Official Review · Reviewer_d2h5 · 2024-07-12

**Soundness:** 3
**Presentation:** 3
**Contribution:** 2
**Rating:** 4
**Confidence:** 4

**Summary:**

This paper proposes an EarlyLate curriculum learner, which distills the easiest samples first and gradually add harder samples. Based on batch-to-global distillation algorithms, the proposed method consistently enhances the distillation performance.

**Strengths:**

- The writing is clear.
- The proposed curriculum learning scheduler seems effective, which is also an interesting point to analyze.
- Good distillation performance.

**Weaknesses:**

Limited contribution and potential overclaiming:

1. In section 3, the initialization with real samples is common in DD, the data selection is proposed by RDED, and only the training scheduler is proposed in this paper. I suggest that, at least, add some diversity analysis and comparison of the distilled data.
2. Though the paper is titled with "diversity-driven", the method part lacks justification of the relation between "diversity" and the proposed scheduler. It seems that only the real initialization contributes to the diversity.

**Questions:**

1. Equation 7: I assume that the initialization samples for $\{x_0, x_1, ...\}$ are ordered by the same criteria in Line 168-170, which is not clarified. The sample order here is critical since $x_0$ will be trained for more iterations than $x_{Mk−1}$. A comparison between random order and ascending/descending order by patch probability could rationalize the model design.
2. Table 3(c) is the most important experiment to support the motivation of the whole paper. Maybe the authors could rewrite the story in the future version by: finding these observations -> thorough analysis -> proposing the method. The current table is not comprehensive enough, and more analysis is appreciated:
    1. Comparison of 1K/1K is non-trivial (baseline).
    2. It is interesting that 4K/2K is worse than 2K/1K, which means training longer leads to a performance drop (they have the same two phases, but 4K/2K trains longer in each phase). The same happens to 4K/1K and 2K/500.
3. In Table 5: 4K-iteration is not fair for comparison since original SRe2L only trains for 1K iterations. A similar concern exists in the model design: e.g. if the base algorithm could converge within 500 iterations, DELT becomes trivial since the algorithms consume all training samples in the last 500 iterations (under default setting MI=4K and RI=500).


Minors:
- Line 66, footnote links to Fig.2
- A figure with training time as the x-axis and training sample number as the y-axis could make the curriculum learning clearer and easier to demonstrate the computation time reduction.

**Limitations:**

The authors have adequately discussed the limitations.

---

> ### Author Rebuttal · Authors · 2024-08-07
>
> We appreciate the reviewer's detailed comments and constructive suggestions, and would like to address some key points from our submission that may have been overlooked, which might have led to confusion and concerns to the reviewer.
>
> >W1: Initialization with real samples is common. Suggest adding diversity analysis and comparison of the distilled data.
>
> Thank you for your suggestion. Our initialization method differs from RDED as we use median scores from the teacher model. We do not select the easiest images as RDED because we want to pick the ones with room for information enhancement via gradient updates. We also have credited RDED appropriately and will add further acknowledgment regarding RDED's initialization idea for the DD task in our revision. However, the core of our method is the *EarlyLate* training approach, not the initialization. Even without this initialization, our method significantly outperforms SRe$^2$L and CDA, with improvements of 9.1% and 2.4%. We also have provided a diversity analysis, specifically the intra-class semantic cosine similarity of synthetic images in Figure 2 left of our submission. Our method achieves the lowest similarity, indicating the highest diversity.
>
> >W2: It seems that only the real initialization contributes to the diversity.
>
> Figure 7 (last column, top to bottom) in our submission clearly demonstrates how our *EarlyLate* strategy enhances diversity in *batch-to-global* optimization methods. Please refer to it for details.
>
> We also highlight that our method produces more diverse images than RDED, as demonstrated in Figure 2 left, even though RDED merges 4 patches together initialized from the original dataset. These statistics demonstrate that simple initialization is not as diverse enough as our *EarlyLate* strategy.
>
> >Q1: Equation 7: A comparison between random order and ascending/descending order by patch probability could rationalize model design.
>
> Thanks for the suggestion. In our DELT, we select the N patches with scores around the median from the teacher model, where the score represents the probability of the true class. To order them, we start with the median, and we go back and force expanding the window around the median until we cover the number of IPCs, refer to Figure 5 for details. The rationale is that these patches present a medium difficulty level for the teacher, allowing more potential for information enhancement through distillation gradients while having a good starting point of information. We empirically validate this approach by comparing different strategies in Table 3 (b) of our paper. We provide these analyses here in the below table including the random order.
>
> We present the impact of using different ordering on ImageNet-100 when having the same initialized images, those around the median, as below:
>
> |     Order     |    DELT    |
> |:------------|:---------:|
> |     Random    |    *67.9*  |
> |   Ascending   |    67.2   |
> |   Descending  |    67.7   |
> | Our DELT |  **68.2** |
>
> Furthermore, we also include a comparison of the performance of different initialization strategies based on the order. Unlike the previous table, the initialized images are different:
>
> | Selection Strategy | DELT |
> |:-------------|:-----------:|
> |       Random       | *67.7*   |
> |      Ascending     |      66.9     |
> |     Descending     |      67.3     |
> |        Our DELT        |    **68.2**   |
>
> >Q2: Table 3 (c).
>
> Thank you for the insightful suggestion. Table 3 (c) is used to identify the optimal hyperparameter for the *EarlyLate* interval and the overall training budget. We have included additional results below and will polish this section in our revised paper accordingly.
>
> >Q2-1: Comparison of 1K/1K is non-trivial (baseline).
>
> Thanks for your suggestion. We provide the results of 1K/1K as below:
>
> | Iterations | Round Iterations  |   Round Iterations   |
> |:------:|:-----:|:-----:|
> |            |        500       |   1K  |
> |     1K     |       44.87      | *43.71* |
> |     2K     |       45.61      | 44.40 |
> |     4K     |       **46.42**      | 44.66 |
>
> We highlight that the 1K/1K configuration yields the lowest score because it cannot apply the *EarlyLate* strategy. With only one round (1K/1K = 1), all IPC images are updated for 1K iterations, which fails to enhance diversity as seen in other experiments in the table.
>
> >Q2-2: It is interesting that 4K/2K is worse than 2K/1K.
>
> Thank you for highlighting this point. More round iterations (RI) do not necessarily lead to better performance with the same number of rounds due to initialization factors. However, increasing total iterations while also increasing the number of rounds is beneficial, as it enhances diversity and overall performance, as shown in Table 3 (c).
>
> >Q3: In Table 5: 4K-iteration is not fair for comparison since original SRe2L only trains for 1K iterations.
>
> We appreciate the reviewer's comments. To clarify, the default setting for SRe$^2$L also uses 4K iterations for their final reported models, as described in their paper (Sec 3.2, "Recover Budget"). In our Table 5, as reiterated in the caption, SRe$^2$L and all other methods are trained with 4K iterations using the official code to ensure a strictly fair comparison with our approach.
>
> Additionally, if the base algorithm converges within 500 iterations, DELT would become 500/100 or 500/250, still significantly faster than the base algorithm. **The mechanism and design of our DELT ensure it will always be much faster than the base method, regardless of the number of iterations needed for convergence.**
>
> Minors:
>
> >Line 66, footnote links to Fig.2
>
> Thank you for pointing this out. We have double-checked the footnote to ensure it references the correct position.
>
> >A figure with training time as x-axis and training sample number as y-axis.
>
> Thank you for the insightful comment. Following the suggestion, we have included a figure (Figure 3) in the PDF attachment to directly illustrate the reduction in computation time.

---

> > ### Comment · Reviewer_d2h5 · 2024-08-11
> >
> > Thanks for the authors' response and some of my concerns are addressed.
> >
> > **W1-W2**: Thanks for the clarification but some concerns on W2 still remain. The authors give empirical results on enhancing diversity, but should clarify why the EarlyLate training enhances diversity. I understand that the method could learn samples at diverse difficulty levels, but there is a gap to pixel diversity. Perhaps the diversity all comes from the initialization.
> >
> > **Q1**: Surprisingly this is not detailed in the paper.
> >
> > **Q2-Q3 and Minors**: Thanks for the details and I appreciate the efficiency analysis.
> >
> > I suggest the authors polish the writing of section 3, also according to reviewers *1XsL* and *cugo*.

---

> > > ### Author Response · Authors · 2024-08-11
> > > **Response to Reviewer d2h5**
> > >
> > > We sincerely appreciate the reviewer's kind reply and further questions.
> > >
> > > >Some concerns on W2 still remain. The authors give empirical results on enhancing diversity, but should clarify why the EarlyLate training enhances diversity. I understand that the method could learn samples at diverse difficulty levels, but there is a gap to pixel diversity. Perhaps the diversity all comes from the initialization.
> > >
> > > Initialization forms the foundation, while the proposed EarlyLate training enhances diversity by varying the optimization length for different samples post-initialization. Without this variation, using the same optimization budget/iteration with the initialization strategy for all samples would result in similar style patterns (e.g., similar mosaics or abstract drawings), as shown in the MTT and SRe$^2$L columns of Figure 7. In our method, fully-optimized images will look like SRe$^2$L/CDA synthesis, under-optimized images will look closer to the original images like RDED/MinimaxDiffusion, and this change is gradual for each class in the entire dataset. If the reviewer examines Figures 6 and the last column of Figure 7 (from top to bottom), they will be seen how our method intuitively increases the diversity of the generated images.
> > >
> > > >Q1: Surprisingly this is not detailed in the paper.
> > >
> > > Given the paper's length limitation and our original belief that Figure 5 was sufficiently informative, we will include the additional details mentioned in our response in the revised version.
> > >
> > > >I suggest the authors polish the writing of section 3, also according to reviewers 1XsL and cugo.
> > >
> > > Thanks for the suggestion. We will polish section 3 carefully according to reviewers 1XsL and cugo's comments.

---

### Official Review · Reviewer_cugo · 2024-07-12

**Soundness:** 3
**Presentation:** 2
**Contribution:** 2
**Rating:** 5
**Confidence:** 4

**Summary:**

Recent advancements in dataset distillation have led to two main approaches: batch-to-batch and batch-to-global matching. While the former excels in small datasets, the latter, though popular for large datasets, faces a diversity challenge due to independent optimization. Authers propose an EarlyLate training scheme that enhances diversity in batch-to-global matching by partitioning IPC samples into subsets and optimizing them locally. Experiments show significant improvements over previous methods.

**Strengths:**

1) The technical approach is solid and robust, demonstrating a high level of technical competence.
2) The performance metrics presented are highly competitive, showcasing the method's effectiveness in comparison to existing benchmarks.

**Weaknesses:**

1) The motivation for the research is unclear, lacking an explicit articulation of the unifying challenges faced by current state-of-the-art works.
2) The resolution of the figures is inadequate, impeding clear interpretation of the results.
3) There is inconsistency in the styling of table borders and captions, with captions for Table 1 and 2 placed in different positions compared to subsequent tables, some above and some below the table.
4) The experimental settings are not uniformly aligned, and efforts should be made to cover all datasets and settings consistently across all experiments to ensure comparability and rigor.

**Questions:**

see weaknesses

**Limitations:**

see weaknesses

---

> ### Author Rebuttal · Authors · 2024-08-06
>
> We appreciate the reviewer's valuable and constructive commetns. We will accommodate all of the suggestions in our revision. In the following, we make further clarifications to reviewer's concerns.
>
> >Q1: The motivation for the research is unclear, lacking an explicit articulation of the unifying challenges faced by current state-of-the-art works.
>
> Thanks for the suggestion. Our proposed DELT method is motivated to address the widely-recognized less-diverse data generation issue in *batch-to-global* methods, which is our primary goal of this work. As we have briefly introduced in the Abstract section, prior state-of-the-art dataset distillation methods like SRe$^2$L (NeurIPS'23) employ *batch-to-global* optimization, where target images are generated by independently optimizing samples using shared global supervision signals across different synthetic images, which suffers limited supervision and synthesis. G-VBSM (CVPR'24) improves matching precision by utilizing a diverse set of signals from multiple backbones and statistical metrics, but the increased model diversity also adds to the overall complexity of the framework, reducing its conciseness. RDED (CVPR'24) uses a train-free image stitching method that crops original images into patches ranked by realism scores from an observer model, but it does not enhance or optimize the visual content within the distilled dataset. Consequently, the diversity and richness of information depend heavily on the original dataset's distribution. That is why RDED does not improve the diversity much (refer to Figure 2 left). Our *EarlyLate* optimization-based solution enjoys the flexibility to fine-tune images with efficient training, offering the advantages of both strategies.
>
> >Q2: The resolution of the figures is inadequate, impeding clear interpretation of the results.
>
> We have included a higher-resolution version of Figure 1 in the rebuttal PDF, please check it out. Datasets like CIFAR and Tiny-ImageNet have low original resolutions (32 $\times$ 32 in Figure 11 and 64 $\times$ 64 in Figure 9), so images may appear blurry when enlarged. We will aim to provide figures with the highest possible resolution in our revised paper.
>
> >Q3: There is inconsistency in the styling of table borders and captions, with captions for Table 1 and 2 placed in different positions compared to subsequent tables, some above and some below the table.
>
> Thanks for pointing this out. We have made the caption location consistent in our manuscript and will update into our revised submission.
>
> >Q4: The experimental settings are not uniformly aligned, and efforts should be made to cover all datasets and settings consistently across all experiments to ensure comparability and rigor.
>
> Thanks for raising this concern. We ensure that all experimental settings across our tables are fully consistent across different methods. This alignment has been thoroughly double-checked during the rebuttal. We are confident that the superior performance of our method is entirely based on the fair comparisons and our proposed approach. If the reviewer has further concerns about the specific table, kindly point them out and we are happy to clarify.

---

> > ### Comment · Reviewer_cugo · 2024-08-12
> >
> > For Q4, plz check table 2, why CIFAR-10 is not in the list while it is mentioned in the caption?

---

> ### Author Response · Authors · 2024-08-12
> **Thanks for pointing this out**
>
> Thank you for pointing this out! The mistake in the caption was unintentional, likely due to following other works. We will remove it in the revision to ensure it aligns with our table.

---

### Official Review · Reviewer_1XsL · 2024-07-13

**Soundness:** 2
**Presentation:** 1
**Contribution:** 2
**Rating:** 5
**Confidence:** 4

**Summary:**

This work studies batch-to-global dataset distillation, optimizing the synthetic dataset by matching the statistical information of the synthetic batches to that of the full real dataset. Previous batch-to-global methods lacked diversity because each batch had the same optimization objective, leading to redundant information being learned across different batches. Based on this, the paper proposes an early-late training method. First, the real data is divided into lowest, medium, or highest probability patches based on a pretrained model, and these patches are sampled to initialize the synthetic dataset. During training, within-class samples are divided into smaller sub-batches, which are gradually concatenated for batch-to-global training.

**Strengths:**

1. Previous batch-to-global methods indeed faced the problem of synthetic datasets receiving the same supervision signal, leading to redundant information being learned. This paper attempts to propose a new solution to this issue.

2. Extensive experiments demonstrate the good efficacy of the proposed method.

**Weaknesses:**

##  The main problem of this work is the writing.

>*It dedicates too much space to introducing previous work.*

The introduction describes previous methods in too much detail, leading to redundancy with the content in the related work section. The related work section also spends too much space summarizing and describing previous methods.

>*The technical part is confused*

The proposed method appears straightforward, but the authors describe the entire process almost entirely in text, lacking mathematical descriptions and definitions, which makes it somewhat difficult to understand. I suggest the authors dedicate more space to explaining the Concatenation Training and Training Procedure, incorporating some formulas to clearly demonstrate how the training is conducted.

>*It is doubtful whether the proposed method can effectively solve the issues present in previous approaches.*

Although the synthetic dataset is further divided within classes and different initializations are used, the supervision signal for each sub-batch seems to still be the same global signal as in other batch-to-global methods. This means that each sub-batch is still optimized in the same direction, potentially resulting in redundant information being learned. I suggest that the authors try to provide a more sound explanation.

**Questions:**

The Training Procedure in Section 3 is somewhat difficult to understand. When training the later sub-batches in DELT, are the previous parts frozen, or are they trained together?

---

> ### Author Rebuttal · Authors · 2024-08-07
>
> We thank the reviewer for the constructive comments. We will accommodate all the suggestions in our revision. In the following, we make further clarifications to reviewer's concerns.
>
> >Q1: Introduction describes previous methods in too much detail.
>
> We appreciate the constructive suggestion. We will streamline the introduction, refine the related work section to merge overlaps in the introduction and focus on providing a concise overview of previous works in the revision.
>
> >Q2: dedicate more space to explaining Concatenation Training and Training Procedure.
>
> Thanks for the suggestion. We explain the detailed process as follows, which will be included in our revision. The representative *batch-to-global* DD methods like SRe$^2$L, CDA, G-VBSM, and our method contain three stages: 1) Pretrain/Squeeze model, the objective of this stage is to extract crucial information from the original dataset. 2) Data synthesis, this phase involves reconstructing the retained information back into the image space utilizing class labels, regularization terms, and BN trajectory alignment. 3) Post-training on synthetic data and evaluation using soft labels. We elaborate on each stage in detail using formulas as follows:
>
> 1) Pretrain/Squeeze Model:
>
> The learning process can be simplified to regular model training on the original dataset using an appropriate training recipe:
>
> \begin{equation}
> \boldsymbol{\theta}_{\mathcal{T}}=\underset{\boldsymbol{\theta}}{\arg \min } \mathcal{L}\_{\mathcal{T}}(\boldsymbol{\theta})
> \end{equation}
>
> where ${\mathcal{T}}$ is an original large labeled dataset and we train the model with parameter $\boldsymbol{\theta}$ on it for data synthesis/recovery.  $\mathcal{L}_{\mathcal{T}}(\boldsymbol{\theta})$ typically uses cross-entropy loss as:
>
> \begin{equation}
> \mathcal{L}_{\mathcal{T}}(\boldsymbol{\theta})=\mathbb{E}\_{(\boldsymbol{x}, \boldsymbol{y}) \in \mathcal{T}}[\boldsymbol{y} \log (\boldsymbol{p}(\boldsymbol{x}))]
> \end{equation}
>
> 2) Data synthesis:
>
> Our *EarlyLate* Optimization:
>
>  \begin{equation}
> \mathrm{Round \ 1}:  \underset{\mathcal{C}\_{\mathrm{IPC}\_{0:k-1}},|\mathcal{C}|}{\arg \min } \ell\left(\phi\_{\boldsymbol{\theta}\_{\mathcal{T}}}\left(\widetilde{\boldsymbol{x}}\_{\mathrm{IPC}\_{0:k-1}}\right), \boldsymbol{y}\right)+\mathcal{R}\_{\text {reg }}
> \end{equation}
>
> ...
>
> \begin{equation}
> \mathrm{Round \ M-1}: \underset{\mathcal{C}\_{\mathrm{IPC}\_{0:Mk-1}},|\mathcal{C}|}{\arg \min } \ell\left(\phi\_{\boldsymbol{\theta}\_{\mathcal{T}}}\left(\widetilde{\boldsymbol{x}}\_{\mathrm{IPC}\_{0:Mk-1}}\right), \boldsymbol{y}\right)+\mathcal{R}\_{\text {reg }}
> \end{equation}
>
>    where $\mathcal{C}$ is the target small distilled dataset with $\widetilde{\boldsymbol{x}}$. $\mathrm{M}$ is the number of batches and $\mathrm{M}>1$ (If $\mathrm{M}=1$, training will degenerate into a way without *EarlyLate*). This process is referred to in Figure 4 of our submission. $\mathcal{R}_{\text {reg }}$ is the regularization term, we also utilize the BatchNorm distribution regularization term to improve the quality of generated images:
>
>  \begin{equation}
> \mathcal{R}\_{\mathrm{reg}}(\widetilde{\boldsymbol{x}})  = \sum_l\left\|\mu_l(\widetilde{\boldsymbol{x}})-\mathbf{B N}_l^{\mathrm{RM}}\right\|_2+\sum_l\left\|\sigma_l^2(\widetilde{\boldsymbol{x}})-\mathbf{B N}_l^{\mathrm{RV}}\right\|_2
> \end{equation}
>
> where $l$ is the index of BN layer, $\mu_l(\widetilde{\boldsymbol{x}})$ and $\sigma_l^2(\widetilde{\boldsymbol{x}})$ are mean and variance. $\mathbf{B N}_l^{\mathrm{RM}}$ and $\mathbf{B} \mathbf{N}_l^{\mathrm{RV}}$ are running mean and running variance in pre-trained model at $l$-th layer, which are globally counted.
>
> 3) Post-training on synthetic data and evaluation:
>
> \begin{equation}
> \widetilde{\boldsymbol{y}}_i=\phi\_{\boldsymbol{\theta}\_{\mathcal{T}}}\left(\widetilde{\boldsymbol{x}}\_{\mathbf{R}_i}\right)
> \end{equation}
>
> where $\widetilde{\boldsymbol{x}}\_{\mathbf{R}\_i}$ is the $i$-th crop in the synthetic image and $\widetilde{\boldsymbol{y}}\_i$ is the corresponding soft label. Finally, we can train the model using the following objective:
>
> \begin{equation}
> \mathcal{L}_{\text {syn }}=-\sum_i \widetilde{\boldsymbol{y}}_i \log \phi\_{\boldsymbol{\theta}\_{\mathcal{C}\_{\text {syn }}}}\left(\widetilde{\boldsymbol{x}}\_{\mathbf{R}_i}\right)
> \end{equation}
>
> Regarding **concatenation training**, we elaborate:
>
> - Our *EarlyLate* tries to enhance the diversity of the synthetic data by varying the number of iterations for different IPCs during data synthesis pahse.
> - This means the first IPC can be recovered for 4K iterations while the last IPC will only be recovered using 500 iterations.
> - To make this process efficient, we share the recovery time (on the GPU) across the different IPCs via concatenation to minimize the time as much as possible.
> - Therefore the first image IPC will start recovery for a couple of iterations, and when it completes iteration 3,500 the last IPC will join it in the recovery phase to get its 500 iterations.
>
> >Q3: A more sound explanation.
>
> Thank you for the suggestion. We have included an illustration in the attached PDF to explain our method from an optimization perspective. Following [1], we visualize the optimization trajectory of the network loss landscape using the same supervision and training data but with different training budgets of 50, 100, 150, and 200 steps (similar to our *EarlyLate* training). This demonstrates that each sub-batch is optimized in different directions, even though the supervision signal appears to be the same global signal used in other batch-to-global methods.
>
> [1] Li, et al. "Visualizing the loss landscape of neural nets".
>
> >Q4: When training the later sub-batches in DELT, are previous parts frozen, or are they trained together?
>
> Thank you for raising this suggestion. In DELT, later sub-batches join the previous sub-batches in recovery/training, rather than freezing the earlier sub-batches. We will clarify this in our revision.

---

> > ### Comment · Reviewer_1XsL · 2024-08-12
> >
> > I thank for the detailed rebuttal. Some concerns are addressed. Nevertheless, I think a major revision would be important so I maintain my initial score.

---

> > > ### Author Response · Authors · 2024-08-12
> > > **Thank you for your post-response**
> > >
> > > Thank you for your post-response, we are glad that some of your concerns are addressed by our rebuttal, and we respect your comments and decision.
> > >
> > > However, we would like to further clarify two points (primarily for the ACs):
> > >
> > > 1. The perception of whether our writing is understandable largely depends on the reader's background. For example, another reviewer mentioned that our writing is clear. The difficulty in understanding may stem from a lack of familiarity with recent developments in *batch-to-global* matching in large-scale dataset distillation methods and related works like SRe$^2$L, CDA, RDED, etc.
> > >
> > > 2. The three stages mentioned in our rebuttal are well-established in recent literature. In our paper, we focused on highlighting the differences and our contributions relative to previous works, instead of reiterating well-known frameworks.
> > >
> > > It seems that the subjective perception of writing clarity was a key concern for this reviewer, while we have proposed revisions to address this, we are unsure if this alone should be a determining factor in the rejection.
> > >
> > > We hope the ACs will consider these points in the final decision.

---

> > > > ### Comment · Reviewer_1XsL · 2024-08-12
> > > >
> > > > I want to clarify that I have the background of batch-to-global matching, e.g., Sre$^2$L, but the initial draft gave me the impression that too much space was devoted to unnecessary parts, while the original content lacked detail. As a result, the paper feels unclear when reading.
> > > >
> > > > However,  I wouldn't be too harsh on this. I am not strongly against accepting the paper, although, a major revision would be necessary. I increase my score to 5.

---

> ### Author Response · Authors · 2024-08-12
> **Thanks for raising the score**
>
> Thank you for your kind words and for raising the score. We have no intention of complaining but are thankful and sincerely appreciate the time and effort you invested in reviewing our work, and further helping us improve it. We assure that we will polish our paper carefully according to the suggested comments. Wishing you a great day!

---

### Author Rebuttal · Authors · 2024-08-06

We appreciate all reviewers for their positive comments, e.g., this paper attempts to propose a new solution to the issue of previous batch-to-global methods indeed faced the problem of synthetic datasets receiving the same supervision signal, leading to redundant information being learned, this is a simple but novel approach to increase the diversity of distilled datasets, I believe it can provide some inspiration for future work [**1XsL, 1mgP**], the writing is clear [**d2h5**], extensive experiments demonstrate the good efficacy of the proposed method, the proposed curriculum learning scheduler seems effective, which is also an interesting point to analyze, the method or the training scheme reduces the computational load compared to batch-to-global matching methods [**1XsL, d2h5, 1mgP**], the technical approach is solid and robust, demonstrating a high level of technical competence, the experiments are comprehensive [**cugo, 1mgP**], the performance metrics presented are highly competitive, good distillation performance [**cugo, d2h5**].

We also appreciate the constructive suggestions, e.g.,  too much space to introduce previous work [**1XsL**], motivation is lacking an explicit articulation [**cugo**], adding some diversity analysis and comparison of the distilled data [**d2h5**], performance of other sota methods on MobileNet-v2 [**1mgP**], etc., which will definitely help us improve the quality of this paper. We will accommodate all of the comments in our revision. Below, we summarize our responses and make further clarifications to questions from each reviewer.

We summarize our rebuttal as follows:

1. We have dedicated more space to explaining the concatenation training and training procedure, incorporating formulas to clearly demonstrate how the training is conducted. [**1XsL**]

2. We have provided a more sound explanation through optimization direction under different training steps in loss landscape to demonstrate how our *EarlyLate* training avoids learning redundant information. [**1XsL**]

3. We have provided more descriptions of the motivation for the research with an explicit articulation of the unifying challenges faced by current state-of-the-art works. [**cugo**]

4. We have provided the higher resolution figures and fixed the inconsistency in the styling of table borders and captions. [**cugo**]

5. We have ensured that the experimental settings are uniformly aligned, and to cover all datasets and settings consistently across all experiments to ensure comparability and rigor. [**cugo**]

6. We have provided a comparison between random order and ascending/descending order by patch probability to rationalize the model design. [**d2h5**]

7. We have provided the comparison of 1K/1K as an additional baseline in the rebuttal and in our revision. [**d2h5**]

8. We have provided additional performance of other sota method RDED on MobileNet-v2 for a more comprehensive comparison. [**1mgP**]

9. We have provided more ablation results with and without initialization/*EarlyLate* to study the impact of the initialization method. [**1mgP**]

We further clarify that:

1. As the synthesis will be updated during training, the initialization will contribute to the final performance but not contribute to the diversity a lot like the proposed *EarlyLate* optimization.

2. We have included the 1K/1K comparison in our rebuttal. It was not in the original submission because this setting does not support *EarlyLate* training and defaults to base framework+initialization.

3. In Table 5: 4K-iteration is for all methods and the original SRe$^2$L also uses 4K for optimization.

We also would like to highlight that:

1. This is the first work to introduce *EarlyLate* training for generating more diverse synthetic images in dataset distillation.

2. Our approach achieves the current best accuracy to different date volumes on both small-scale and large-scale datasets, surpassing all previous state-of-the-art methods by significant margins.

---

### Decision · Program_Chairs · 2024-09-25

**Decision:**

Reject

**Comment:**

This submission presents a new training approach for dataset distillation from the perspective of enhancing the diversity of synthetic images per class. The proposed approach first selects patches from real images to initialize synthetic images, and then optimizes the generation process for the current batch by reusing synthetic images from all preceding batches and introducing new batched images, which is called the early-late training scheme. The submission was finally scored (5 yet less positive,5,4,5) by four knowledgeable reviewers, who mostly acknowledged the simplicity as well as the good performance of the proposed method, but raised several major concerns about **1)** incremental technical novelty, especially to previous methods; **2)** poor presentation of the method section; **3)** unclear motivation/insight of the real data selection criteria in the initialization step; **4)** questionable core contribution, that is, performance improvement is mostly from the initialization step rather than the early-late training scheme; **5)** inconsistent experimental settings in performance comparison.

The authors provided detailed responses to these concerns, and all reviewers acknowledged that some of their concerns are addressed, but three of them did not increase their scores after the rebuttal. The AC read the paper, the reviews, the rebuttal and the reviewers' feedback. I also think the proposed method is simple and shows good performance, but lacks strong technical novelty (using real data for synthetic data initialization and the concept of early-late training are both not new in dataset distillation research [15,27,28,12]) or any theoretical principles in the method formulation. Besides, the experiments provided in the original manuscript and in the rebuttal show that the initialization step clearly outweighs the early-late training (claimed core contribution) in terms of final performance improvement, though the method/criteria used in the initialization step is common. The early-late training brings small gain (sometimes less than 1%, at most 1.5%) to the initialization step. Therefore, this submission at its current form does not meet the bar for acceptance. The authors are encouraged to consider the reviewers' comments and suggestions to improve their work for a future conference.